# Combining Fixed-Time Insemination and Improved Catheter Design in an Effort to Improve Swine Reproduction Efficiency

**DOI:** 10.3390/ani9100748

**Published:** 2019-09-29

**Authors:** Matthew McBride, Rocio Amezcua, Glen Cassar, Terri O’Sullivan, Robert Friendship

**Affiliations:** Department of Population Medicine, University of Guelph, Guelph, ON N1G2W1, Canada; mmcbri02@uoguelph.ca (M.M.); mamezcua@uoguelph.ca (R.A.); tosulliv@uoguelph.ca (T.O.); rfriends@ovc.uoguelph.ca (R.F.)

**Keywords:** insemination rods, fixed-time insemination, luteinizing hormone, equine chorionic gonadotropin

## Abstract

**Simple Summary:**

Conventional practice is to breed sows by artificial insemination (AI) at least twice using approximately three billion sperm per insemination. Sows may be bred only once using the technique of fixed-time artificial insemination (FTAI) if ovulation is predictable. This research explored the use of combining a single fixed-time artificial insemination (FTAI) and an alternative insemination catheter design that reportedly reduces semen backflow in order to reduce semen dosage and maintain reproduction efficiency. The FTAI technique used in this study involved two hormone treatments, 80 h apart, after weaning followed by a single insemination. The two catheters used in this study were a conventional foam-tipped insemination catheter and a Gedis catheter, which is designed to be completely inserted into the vagina of the sow. The semen is enclosed along the length of the catheter and held in place by a gel cap that melts when inserted into the cervix. Sows were assigned to the following treatments: Group 1 (*n* = 135), bred twice with a conventional catheter and a standard semen dose of approximately three billion sperm in 80 mL; Group 2 (*n* = 123), FTAI with conventional catheter and a standard semen dose; Group 3 (*n* = 127), FTAI with Gedis catheter and a standard semen dose; Group 4 (*n* = 126), FTAI with Gedis catheter and a reduced semen dose with approximately one billion sperm. The farrowing rates were 81.6%, 77.7%, 74.0%, and 62.7% for Groups 1 to 4, respectively. Litter sizes of Group 3 and Group 4 were smaller than Group 1. Overall, the combination of Gedis catheter and FTAI resulted in decreased reproductive performance that outweighed the value of using less semen.

**Abstract:**

Conventional practice is to breed sows by artificial insemination (AI) at least twice using approximately three billion sperm per insemination upon estrus at standing heat. This research explored the use of combined technologies, including fixed-time insemination (FTAI) and an alternative catheter design that reportedly reduces semen backflow, in order to reduce the number of inseminations and the semen dosage and maintain reproductive efficiency. The FTAI technique used in this study was to inject I.M. 600 IU equine chorionic gonadotropin (eCG) at weaning and 5 mg porcine luteinizing hormone (pLH) to stimulate ovulation 80 h later, followed by a single insemination 36 h after the pLH injection. The two catheters used in this study were a conventional foam-tipped insemination catheter and a Gedis catheter. The Gedis catheter is designed to be completely inserted into the vagina. The semen is enclosed along the length of the rod and held in place by a gel cap that melts when inserted into the cervix. Sows were assigned to the following treatments: Group 1 (*n* = 135), bred twice with a conventional catheter and a standard semen dose of approximately three billion sperm in 80 mL; Group 2 (*n* = 123), FTAI with conventional catheter and a standard semen dose; Group 3 (*n* = 127), FTAI with Gedis catheter and a standard semen dose; Group 4 (*n* = 126), FTAI with Gedis catheter and a reduced semen dose with one billion sperm. The farrowing rates were 81.6%, 77.7%, 74.0%, and 62.7% for Groups 1 to 4, respectively. The likelihood of farrowing was lower for Group 3 and Group 4 compared to Group 1 (odds ratio (OR) = 0.57; *p* = 0.08 and OR = 0.35; *p* = 0.001, respectively). Likewise, litter size of Group 3 and Group 4 was smaller than Group 1 (*p* = 0.006 and *p* = 0.04, respectively). Overall, the combination of Gedis catheter and FTAI resulted in decreased reproductive performance that outweighed the value of using less semen.

## 1. Introduction

Conventional practice is to breed sows by artificial insemination (AI) at least twice using approximately three billion sperm per insemination upon estrus at standing heat. Single, fixed-time artificial insemination (FTAI) of weaned sows can result in reproductive performance that is comparable to traditional multiple inseminations [1,2,3,4]. FTAI can be done following stimulation of ovulation, or by synchronizing both estrus and ovulation in weaned sows and gilts. Significant labour savings can be achieved as well as a narrower window of farrowing time resulting in more uniform-sized piglets at weaning. In a previous study it was shown that the use of equine chorionic gonadotropin (eCG) at weaning induced estrus and the injection of porcine luteinizing hormone (pLH) 3–4 days later resulted in an ovulation about 38 h after pLH [1]. The high degree of predictability resulted in better pregnancy and farrowing rates in treated sows/gilts with a single FTAI than control sows that were inseminated twice.

Intra-cervical insemination is the most common AI technique used on commercial farms [5]. There are a variety of semen catheter designs that allow semen to be deposited into the cervix. Examples of different catheter styles used are the IMV Technologies’^®^ (Haryana, India) foam-tipped catheter (Goldenpig^®^, Hopland, CA, USA) [6]; the Melrose catheter, which is made of a spiral rubber hybrid material which can be washed and reused [6]; and the disposable spirette catheter (SafeBlue^®^ from Minitube, Ingersoll, ON, Canada), which is a modern advancement of the Melrose catheter design. The Gedis^®^ (IMV Technologies^®^) catheter is the only catheter with the semen enclosed along the length of the catheter and held in place by a heat-sensitive gel plug, which liquefies due to the natural body temperature of the sow, allowing semen to be drawn through the cervix and into the uterus [6,7]. It is claimed that the person doing the insemination needs only to insert the device and leave it in place as they move to the next sow, which simplifies insemination and reduces the time of breeding if large numbers of sows are being inseminated. It is also claimed that this method allows for better hygiene standards and a reduction in the volume of semen used per breeding because backflow is decreased [6,7].

The main goal of AI is to ensure that an adequate population of spermatozoa reaches the site of fertilization during the pre-ovulatory period to achieve successful fertilization of all the ova [8]. Standard AI protocols recommend the use of two to five billion sperm cells in 80 to 100 mL volume of extender, repeated once or twice during estrus, so a total of 4 to 12 billion sperm cells may be used per female in each estrus [5,9,10,11]. These conditions limit the number of sows that can be bred from one ejaculate. There is obviously an advantage to increasing the number of sows that a boar with superior genetics can breed. The two basic ways this can be achieved are to firstly, utilize schemes which reduce the number of inseminations per estrus and secondly, to reduce the concentration of spermatozoa per insemination [12]. While unsatisfactory results have been seen when AI dose is reduced below 2.0 billion spermatozoa [9], it has been suggested that one billion spermatozoa per AI dose will give good fertility as long as the timing of insemination is optimal and no backflow of semen occurs [13,14]. If the time of ovulation is known, it is possible to inseminate once at the appropriate time and use less semen per insemination [1,15]. However, results of using lower doses of semen are inconsistent and more research is required to determine the optimum dosage of sperm, the best type of catheter, and the most effective method to control ovulation.

The objective of the present study was to determine whether a combination of techniques including the type of catheter used in insemination (Gedis vs. conventional foam-tipped) and a hormonal protocol (eCG followed by pLH) to induce ovulation can allow the use of a single FTAI with reduced sperm numbers (1 billion vs. 3 billion) for the insemination of sows to achieve reproductive performance comparable to the industry norm achieved with double mating using approximately 3 billion sperm per insemination.

## 2. Materials and Methods

This research trial was approved by the Animal Care Committee of the University of Guelph (#3532), in accordance with the guidelines set forward by the Canadian Council of Animal Care.

### 2.1. Pigs and Treatment Groups

This experiment was conducted at the Arkell Swine Research Station, University of Guelph. Systematic random sampling was used to assign 511 sows at weaning to one of four treatment groups. The Arkell herd has approximately 300 breeding sows and utilizes a monthly batch farrowing system.

● Group 1—Control group (*n* = 135)

Following weaning (Day 0), estrus detection with fenceline boar exposure was performed for approximately 20 min, twice per day beginning on Day 3. Sows exhibiting estrus were bred by AI when first discovered in strong standing heat, and a second insemination was performed 24 h later if the sow was still in standing heat. The catheter used in this breeding was a conventional foam-tipped type, and the semen used was a standard dose of extended semen containing approximately three billion sperm in 80 mL of extended semen.

● Group 2—FTAI using foam-tipped catheter (*n* = 123)

Following weaning (Day 0), sows were injected intramuscularly (IM) with 600 IU of equine chorionic gonadotrophin (eCG, Pregnecol^®^, Vetoquinol, Lavaltrie, QC, Canada) and 80 h later were injected IM with 5 mg of porcine luteinizing hormone (pLH, Lutropin^®^, Vetoquinol, Lavaltrie, QC, Canada). Thirty-six hours after receiving the pLH injection, a single insemination with a standard semen dose was performed using a foam-tipped catheter.

● Group 3—FTAI using Gedis catheter (*n* = 127)

Following weaning (Day 0), sows were injected intramuscularly (IM) with 600 IU of eCG and 80 h later were injected IM with 5 mg of pLH. A single insemination with a standard semen dose was performed 36 h after receiving pLH using a Gedis catheter.

● Group 4—FTAI using Gedis catheter and reduced sperm (*n* = 126)

Same procedure as Group 3 but with a reduced AI dose of one billion sperm in 80 mL of extended semen.

Following insemination, sows were housed in stalls until pregnancy detection, after which they were held in groups until they entered the farrowing room. Estrus detection was done using fenceline boar contact and all inseminations were done in the presence of a boar. The same boar was used for all treatment groups. Only sows displaying estrus were bred. The same MOFA™ extender (AndroPRO^®^ Plus) was used for both the Gedis and regular tubes of semen. Pregnancy detection was done via transabdominal ultrasonography (SonopTek wireless ultrasound scanner SV-1, SonopTek Ltd., Madrid, Spain). This procedure was carried out approximately 28 days after breeding. Sows found not pregnant on the first check were re-checked on Day 35.

Data recorded included: Sow identification number, parity at breeding, treatment, batch number, treatment dates, breeding dates, pregnancy confirmation, farrowing date, room and pen in the farrowing area, piglet birth weights, total number of piglets born (including still births and mummies), and total born alive. The total litter weight was derived by adding the birth weights of all piglets born alive in each corresponding litter. To determine if there was a difference in improvement of breeding between the earlier and later portions of the study, replicates were categorized as: Batches (1 to 6) and Batches (7 to 12).

### 2.2. Data and Statistical Analysis

Data were analyzed using Stata (Stata/SE 14.2 for Mac; StataCorp, College Station, TX, USA). Descriptive statistics including means, standard deviations, and proportions were calculated. Univariate analysis using chi-square test was used to determine associations between categorical variables with treatment groups. A one-way ANOVA was used to determine the difference among treatment groups and continuous variables. In cases where continuous variables were not normally distributed, Kruskal–Wallis test was used to test for significant differences among treatment groups.

The differences in the likelihood of farrowing were analyzed using a logistic model. Results were expressed as odds ratios (OR). Litter size (total born) was analyzed using a mixed linear model. The effect of treatment and parity (at time of insemination) were tested in each model. In the mixed linear model, batch was modeled as a random effect. Significance was determined at *p* < 0.05. When *p* values were between 0.05 to 0.1, they were reported as trends.

## 3. Results

Ten sows in Group 1 and two sows in Group 2 were not bred because the sows did not show signs of estrus. All sows in Group 3 and 4 were inseminated. All bred sows in Group 1were inseminated twice, as intended. The mean time from insemination to farrowing was 115.9 (standard deviation (SD) = 1.3, min = 112, max = 119) days and did not differ among treatment groups. The descriptive statistics of all production parameters are included in Table 1. Group 4 pregnancy rate was significantly lower compared to all other groups. Pregnancy rate was lower in Group 3 compared to Group 1 (*p* = 0.002). Farrowing rate was also lower for sows in Group 4 compared to all other groups. Litter size (both total born and number born alive) tended to be lower for sows in Group 3 compared to Group 1 (*p* = 0.1 and *p* = 0.06, respectively). The average weight of pigs born alive was significantly lower in Group 1 litters compared to Group 3 and Group 4 (*p* < 0.05).

Parity of sows at breeding was even across treatments and ranged from 1 to 9. Parity was categorized as follows for analysis: Parity 1 (118 sows); Parity 2–3 (204 sows) and Parity >3 (179 sows). Parity was not recorded for 10 sows. There was no significant difference in parity among treatment groups (*p* = 0.2) (Table 1). There was no effect of replicate during the trial.

Group 3 sows and Group 4 sows had a lower likelihood of farrowing than Group 1 sows (OR = 0.57, *p* = 0.08, and OR = 0.33, *p* = 0.001, respectively). However, the likelihood of farrowing did not differ between Group 1 and Group 2 sows. Group 4 sows had a lower likelihood of farrowing compared to Group 2 sows (OR = 0.49, *p* = 0.01) and Group 3 sows (OR = 0.61, *p* = 0.08). Group 3 had a lower likelihood of farrowing than Group 2 (OR = 0.6, *p* = 0.08) (Table 2).

Group 3 sows had 1.6 fewer pigs in their litters than sows in Group 1 (*p* = 0.006). Sows in Group 4 had 1.2 fewer pigs in their litters than sows in Group 1 (*p* = 0.04). Group 2 did not differ from the controls (Group 1). Group 2 sows had 1.2 more pigs born than Group 3 but did not differ with Group 4. No significant differences were observed between Group 3 and Group 4. Parity >3 sows had 2.3 more pigs (*p* < 0.001) than sows that were parity 1, and 1.41 more pigs in their litters than sows at parity 2–3 at breeding (*p* = 0.003) (Table 3).

## 4. Discussion

The combined effect of using the Gedis catheter and FTAI did not provide superior reproductive performance compared to double mating with the foam-tipped catheter. Treatment with eCG-pLH to induce ovulation for a FTAI protocol resulted in better reproductive performance for the foam-tipped catheter group than the Gedis group at the same dose of 3 billion sperm. Further reducing the sperm dose from three billion to one billion with the Gedis catheter resulted in lower farrowing rates and litter sizes. The combination of FTAI and Gedis catheter did not make using reduced semen dose possible without some loss in reproductive performance.

The best conception rate, farrowing rate, and largest litter size was achieved by breeding with a conventional foam-tipped semen catheter. This included both Group 1, which was bred when sows were first found in standing heat and rebred 24 h later, and Group 2, using an eCG-pLH FTAI protocol. These results are consistent within the existing literature. One study that utilized a similar FTAI protocol found that reproductive performance was comparable to that of controls with the same semen dose and similar AI catheters [1].

Litter size was significantly smaller in sows bred using the Gedis catheter compared to the control group of sows bred twice using the foam-tipped catheter. This is inconsistent with published literature. Past research indicated a slight improvement in overall reproductive performance with the Gedis catheter, resulting in increased litter sizes compared to controls bred twice via conventional AI [16]. Others reported that breeding multiparous sows with the Gedis catheter at doses of 2.5–3 billion spermatozoa resulted in high reproductive success, with litter sizes of 13.21 [17]. This success may be attributed to the expertise gained from the prolonged use of this style of catheter.

The decline in pregnancy and farrowing rate associated with the reduced semen dose (one billion sperm and the Gedis catheter indicates in part that semen quantity may affect reproductive performance when intracervical insemination is used. Another research group found a remarkable decrease in pregnancy rate and litter size when multiparous sows were inseminated by intracervical means with a dose of three billion (84%, 12.1 pigs per litter) versus two billion sperm (75.8%, 11.7 pigs per litter), respectively [18].

Although thought to benefit reproductive performance, our findings did not indicate the Gedis catheter was superior to the foam-tipped catheter. One limit to this study may have been the lack of familiarity with the Gedis catheter in early stages of the trial. Although there was no significant effect of replicate, there was a trend of improved reproductive performance in sows bred with the Gedis catheter as the trial progressed. This could mean that there may be a period of learning with the introduction of the Gedis catheter, after which comparable reproductive performance may be achieved while using three billion sperm and FTAI regardless of the catheter used. At the beginning of this trial, the breeding technician was unfamiliar with the Gedis catheter and although a training session was conducted prior to the start of the trial this may not have been sufficient and more experience was required.

Utilizing a single, fixed-time insemination combined with the Gedis catheter to achieve pregnancy is one step towards reduced semen use, but more research is required to ensure that desirable farrowing rates and reproductive performance are maintained.

The ability to achieve pregnancy in sows with reduced numbers of sperm per AI dose is advantageous since it can decrease the costs of breeding for the producer. On a larger scale, it would allow for the use of semen from superior boars across a larger population of sows with faster genetic progress in the swine industry. Although this study found that using one billion sperm per insemination resulted in lower farrowing rates and smaller litter size compared to three billion sperm per dose, this is an area of study that needs to be examined further. It may be that with the intracervical insemination techniques used in this study, a dose of semen lower than three billion may not allow sufficient sperm to reach the ampulla for fertilization. Even though the Gedis catheter design acts to reduce semen backflow by acting as a plug with the cervix, a dose of three billion still may not be sufficient to optimize conception. Alternative AI techniques, such as deep-uterine insemination is another method which has been shown to allow reduced sperm dosage and maintain breeding success [19]. The objective of using lower sperm numbers and fewer inseminations to allow wider use of superior boars while maintaining reproductive performance is very worthwhile and more research is required.

## Figures and Tables

**Table 1 animals-09-00748-t001:** Reproduction performance of sows assigned to one of four treatments.

Number of Sows Assigned		Group 1	Group 2	Group 3	Group 4	*p*
		135	123	127	126	
Number inseminated (% inseminated sows)		125 (93.1)	121 (98.4)	127 (100)	126 (100)	
Pregnancy check positive (% positive of sows bred)		108 (86.4) ^a^	101 (83.5) ^ab^	101 (79.5) ^b^	86 (68.2) ^c^	0.002
Farrow (% inseminated sows that farrow)		102 (81.6) ^a^	94 (77.7) ^a^	94 (74) ^a^	79 (62.7) ^b^	0.005
Number of sows within parity categories (%)	1	31 (23.5)	34 (28.1)	30 (23.8)	23 (18.8)	0.2
	2–3	60 (45.4)	50 (41.3)	43 (34.1)	51 (41.8)	
	>3	41 (31)	37 (30.6)	53 (42)	48 (39.3)	
**Mean (SD)**						
Total born per litter		13.4 (3.3) ^a^	13.04 (4.15) ^ab^	12.02 (4.5) ^b^	12.5 (4.4) ^ab^	0.09
Born alive per litter *		11.6 (3.4) ^a^	11.2 (3.4) ^ab^	10.3 (3.9) ^b^	10.9 (3.7) ^ab^	0.08
Total litter birth weights (kg)		17.5 (5.4)	17.01 (4.6)	16.2 (5.6)	17.8 (4.9)	0.1
Piglet birth weight (kg) *		1.53 (0.27) ^a^	1.55 (0.25) ^ab^	1.62 (0.35) ^cb^	1.64 (0.26) ^c^	0.03
Breeding to farrow ** (days)		116.06 (1.4)	115.8 (1.1)	116.03 (1.3)	115.9 (1.2)	0.7

* Kruskal–Wallis test used. ** counted from time of first insemination in Group 1. Group 1 = conventional breeding with insemination of 3 billion sperm with foam-tipped catheter. Group 2 = eCG + pLH 1 insemination of 3 billion sperm with foam-tipped catheter. Group 3 = eCG + pLH 1 insemination of 3 billion sperm using Gedis catheter. Group 4 = eCG + pLH 1 insemination of 1 billion sperm using Gedis catheter (in this and other tables). SD = standard deviation. ^a,b,c^ Different letters means different level of significance.

**Table 2 animals-09-00748-t002:** Effect of treatment and parity at breeding on the likelihood of farrowing.

Fixed Portion	OR ^†^	Lower CI *	Higher CI *	*p*
Group 1	Referent			
Group 2	0.72	0.37	1.38	0.3
Group 3	0.57	0.30	1.07	0.08
Group 4	0.35	0.19	0.65	0.001
Parity 1	Referent			
Parity 2–3	1.28	0.75	2.2	0.3
Parity >3	1.39	0.80	2.4	0.2

† OR = odds ratio. ***** CI = confidence intervals.

**Table 3 animals-09-00748-t003:** Effect of treatment and parity at breeding on total litter size.

Fixed Portion	Coefficient	SE *	Lower CI **	Higher CI **	*p* Value
Group 1	Referent				
Group 2	−0.41	0.57	−1.5	0.71	0.4
Group 3	−1.6	0.57	−2.7	−0.46	0.006
Group 4	−1.2	0.6	−2.4	−0.02	0.04
Parity 1	Referent				
Parity 2–3	0.86	0.56	−0.22	1.9	0.1
Parity >3	2.3	0.55	1.1	3.4	<0.001

* SE = standard error. ****** CI = confidence intervals.

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
