# Peer review of "Combining Fixed-Time Insemination and Improved Catheter Design in an Effort to Improve Swine Reproduction Efficiency"

_animals, 2019, doi:10.3390/ani9100748_

Round 1

Reviewer 1 Report

I have read the manuscript and have some suggestions. 

Overall the paper is sound and the reasoning behind it logical. Sentence structure is good and the paper reads easily. I find it to be a relevant area of research for the pig industry, whose margins are slim (particularly in Australia where I am) and who are constantly searching to reduce costs. It is a good avenue to consider when having to use fresh semen. There are a few comments below. Mainly on the structure and form of the paper. I have not made changes to a PDF copy- but rather laid them out in the following document - line by line.   Overall, my main concerns would be

1) that the methods flow over into the results and they need further separating. The results read strangely because of this.

2) I am a bit concerned with that stats - that when analysing treatment parity and some other fixed effects could have been adjusted for in the model, as the author states that parity has made a difference on variables measured. Therefore, if parity is not even across treatments, not adjusting for this when analysing treatment effects could sway the results. At the VERY least, the reader needs to be ensured that the party was controlled for and even across treatments (if it was), so as not to sway treatment effects with the differences due to parity.

3) there are specific comments below to smaller things which will improve the paper. 

Specific comments:

L15: Not sure if you need the ‘1 of 4 treatments’ . ‘The following treatments’ would do.

L17: I am undecided about the 1-3 billion sperm throughout. I think perhaps in the methods somewhere you need to introduce it and say something along the lines of ‘an estimated 1 billion sperm in …mls and an estimated 3 billion sperm in /….mls.’. Explain what the actual full dose is and what it contains, along with this 1-3 billion sperm.

23:  finish sentence with something like ‘upon oestrus at standing heat’.

27: after luteinizing hormone section say something like ‘to stimulate ovulation’

31: again don’t think you need the ‘1 of 4’

36: has OR been previously defined?

Is there space here to present actual data? We could have litter size and maybe farrowing likelihood data in the abstract?

40: keywords, space for plenty more

45: FTAI – should it be 'stimulation' rather than 'synchronisation' here

I would have the 3 billion before the 1 billion in the brackets as that is the way round that the sentence leading into it discusses them (because you are talking about ‘reduced sperm numbers’). 300, multiparous sows? Further explain boar exposure. Fence-line? 20 min? Same boar for all treatments? For eg Etc ‘if the sow was still standing’ Hoe many sows didn’t get a second insemination? Ie what proportion of sows? This is probably fairly important. If there weren’t any then you don’t need this second bit of the sentence. ‘with 3 billion sperm’ or ‘OF 3 billion sperm’? ‘breedings’ not sure the best term ‘inseminations’? Breeding sounds like live mates ‘and delivered Wednesday’ Relevant to the reader? Make into one sentence. ‘Pregnancy detection was carried out vie …….. on 21-28 days gestation.’ ‘of the sows where pregnancy was not detected’ – bit awkward. Maybe ‘Sows found non-pregnant on the first check were re-checked at week 5’ Probably shouldn’t be wk 5 if the other was referred to as 21-28 days. Would it be 35 days? Was the total number of piglets born including mummies. Probably define in brackets. Ie (Live born, stillborn and mummified foetuses) or ( not including mummified foetuses) I don’t know if I understand the stats completely. Why have they not been analysed so that fixed effects can be adjusted for when analysing treatment? For example parity and farrowing room could be adjusted for when analysing treatment.

Results L 140-144. This section sounds like all methods

in brackets you have .1 and .06 make sure that in all brackets you put 0.1 and 0.06 etc.

156-157. this is methods

160-162. this is methods

remove all above and then at the end of this sentence ‘…….. respectively) (Table 3)’. I don’t think you need to specify the statistical analysis at the start of every results paragraph, it reads strangely. Remove this and shert with ‘Group 3 sows had 1.6 less ….’ And then finish the sentence with ‘(Table 4)’. I think the layout for this paper allows you to imbed the tables where you would like them to be? No? I don’t know if having all of them at the end of the results is right. Read instructions to authors.

Table 1- not sure if ‘bred’ is the right term. Mabye ‘inseminated’. – it is used several times in the same table

Table 1- a and bs need to be superscripted a b

177-182. bring in the text to the left margin – shouldn’t be in middle (all tables)

Table 2. Don’t think the title is stand alone – perhaps expand sightly

Table 2. I am not 100% sure that table 2 is needed. There is a lot of detail and space taken up by something that is not linked to your aims and methods. I think it would be enough to have this in text perhaps.

Table 2: If you are keeping the table in – the N= distracts from the numbers you are presenting and it is a little messy. Can they go somewhere else, like in the text below the table when you describe the groups maybe.

Table 3. Don’t need to state stats used in title heading

Table 3. Do you explain in your methods somewhere why you have chosen to group parity in order to analyse it? Ie why is it not just 1,2,3,4,5,6? Why did you choose these groups to make? \

Table 1 has superscripts but the others done – can they be implemented on 3 and 4 to make the data clearer?

Table 4. ‘Ref’ is shortened. Need to define or write in full.

Discussion : 231. Any suggestion as to the reason behind the differences? Did their methods differ subtly, or genetics, semen quality??

This statement needs explaining further. Why do these results point to this? Is it not just because of numbers of sperm? Why quality? Explain.

241-248 and 251-254. These sections say exactly the same thing. Either needs combining or remove the whole second para and find where to put the information in the first sentence (249+).

References

Are the semicolons with the author names in the reference guide, I don’t remember them?

I believe that at the end of the reference Animal requires ‘Available online: www…. (accessed on ……….).’ for every reference

Reference 2.Extra space at start

Reference 3. Needs lining up with others

Reference 9. Remove hyperlink from title

Ref 11. IS there not more to this one? It seems incomplete?

Ref 13. Everyone else has a bolded year this one doesn’t. Refer to the reference guide for authors as to which way this should be done.

Author Response

I have read the manuscript and have some suggestions. 

Overall the paper is sound and the reasoning behind it logical. Sentence structure is good and the paper reads easily. I find it to be a relevant area of research for the pig industry, whose margins are slim (particularly in Australia where I am) and who are constantly searching to reduce costs. It is a good avenue to consider when having to use fresh semen. There are a few comments below. Mainly on the structure and form of the paper. I have not made changes to a PDF copy- but rather laid them out in the following document - line by line.   Overall, my main concerns would be

1)that the methods flow over into the results and they need further separating. The results read strangely because of this.

This has been addressed and methods that were repeated in the results have been removed

2) I am a bit concerned with that stats - that when analysing treatment parity and some other fixed effects could have been adjusted for in the model, as the author states that parity has made a difference on variables measured. Therefore, if parity is not even across treatments, not adjusting for this when analysing treatment effects could sway the results. At the VERY least, the reader needs to be ensured that the party was controlled for and even across treatments (if it was), so as not to sway treatment effects with the differences due to parity.

Actually Parity was controlled for in both models. Parity was even across treatments and descriptive statistics was added to the Table 1.

3) there are specific comments below to smaller things which will improve the paper. 

Specific comments:

L15: Not sure if you need the ‘1 of 4 treatments’ . ‘The following treatments’ would do.

This suggestion has been accepted and applied.  Changed to ‘the following treatments’

L17: I am undecided about the 1-3 billion sperm throughout. I think perhaps in the methods somewhere you need to introduce it and say something along the lines of ‘an estimated 1 billion sperm in …mls and an estimated 3 billion sperm in /….mls.’. Explain what the actual full dose is and what it contains, along with this 1-3 billion sperm.

This suggestion has been accepted and applied. A standard dose was explained

23:  finish sentence with something like ‘upon oestrus at standing heat’.

This suggestion has been accepted and applied

27: after luteinizing hormone section say something like ‘to stimulate ovulation’

This suggestion has been accepted and applied

31: again don’t think you need the ‘1 of 4’

This suggestion has been accepted and applied.  Changed to ‘the following treatments’

36: has OR been previously defined?

   OR is pretty standard term, however in Data and Statistical analysis a sentence staying that results from the logistic model were expressed as Odd ratios (OR)

Is there space here to present actual data? We could have litter size and maybe farrowing likelihood data in the abstract?

The abstract already exceeds the word limit.  There is no more room for additional data

40: keywords, space for plenty more

We have added 2 more keywords

45: FTAI – should it be 'stimulation' rather than 'synchronisation' here

This suggestion has been accepted and applied.

I would have the 3 billion before the 1 billion in the brackets as that is the way round that the sentence leading into it discusses them (because you are talking about ‘reduced sperm numbers’). 300, multiparous sows? Further explain boar exposure. Fence-line? 20 min? Same boar for all treatments? For eg Etc ‘if the sow was still standing’ Hoe many sows didn’t get a second insemination? Ie what proportion of sows? This is probably fairly important. If there weren’t any then you don’t need this second bit of the sentence. ‘with 3 billion sperm’ or ‘OF 3 billion sperm’? ‘breedings’ not sure the best term ‘inseminations’? Breeding sounds like live mates ‘and delivered Wednesday’ Relevant to the reader? Make into one sentence. ‘Pregnancy detection was carried out vie …….. on 21-28 days gestation.’ ‘of the sows where pregnancy was not detected’ – bit awkward. Maybe ‘Sows found non-pregnant on the first check were re-checked at week 5’ Probably shouldn’t be wk 5 if the other was referred to as 21-28 days. Would it be 35 days? Was the total number of piglets born including mummies. Probably define in brackets. Ie (Live born, stillborn and mummified foetuses) or ( not including mummified foetuses) I don’t know if I understand the stats completely. Why have they not been analysed so that fixed effects can be adjusted for when analysing treatment? For example parity and farrowing room could be adjusted for when analysing treatment.  .

Parity was included in both models (see Table 3 and 4, now tables 2 and 3). Room was not included in the likelihood of farrowing since all sows in those rooms farrowed. For total born there were not big differences on the coefficients in the model when including room. In the simple association there were not sigifniciant differences among the rooms for total born pigs)

We have changed the order of 3 billion and 1 billion.

Any breeding herd would be a mix of multiparous sows and gilts, so we did not include ‘multiparous”

We explained ‘boar exposure’ further.

Changed ‘breedings’ to ‘inseminations’

We have now further explained number of inseminations.

Removed ‘Wednesday’ to ‘delivered weekly’

We have excepted the description of pregnancy detection and have changed it.

We have further explained litter size

Results L 140-144. This section sounds like all methods

Deleted this in Results

in brackets you have .1 and .06 make sure that in all brackets you put 0.1 and 0.06 etc.

This change has been accepted and applied

156-157. this is methods

160-162. this is methods

Deleted this in Results

remove all above and then at the end of this sentence ‘…….. respectively) (Table 3)’. I don’t think you need to specify the statistical analysis at the start of every results paragraph, it reads strangely. Remove this and shert with ‘Group 3 sows had 1.6 less ….’ And then finish the sentence with ‘(Table 4)’. I think the layout for this paper allows you to imbed the tables where you would like them to be? No? I don’t know if having all of them at the end of the results is right. Read instructions to authors.

Suggestions accepted and changes made.

Table 2 has been removed, but the Tables were kept in the same location

Table 1- not sure if ‘bred’ is the right term. Mabye ‘inseminated’. – it is used several times in the same table

bred’ changed to ‘inseminated’

Table 1- a and bs need to be superscripted a b

This change has been made

177-182. bring in the text to the left margin – shouldn’t be in middle (all tables)    

This change has been made

Table 2. Don’t think the title is stand alone – perhaps expand slightly

Table 2 has been removed

Table 2. I am not 100% sure that table 2 is needed. There is a lot of detail and space taken up by something that is not linked to your aims and methods. I think it would be enough to have this in text perhaps.

Table 2 has been removed

Table 2: If you are keeping the table in – the N= distracts from the numbers you are presenting and it is a little messy. Can they go somewhere else, like in the text below the table when you describe the groups maybe.

Table 2 has been removed

Table 3. Don’t need to state stats used in title heading

This change has been made

Table 3. Do you explain in your methods somewhere why you have chosen to group parity in order to analyse it? Ie why is it not just 1,2,3,4,5,6? Why did you choose these groups to make? \

This has now been added to the text in the results section

Table 1 has superscripts but the others done – can they be implemented on 3 and 4 to make the data clearer?

In Table 3 and 4  (now Tables 2 and 3) cannot be included since it is a model. Usually the superscripts are reported only for descriptive data to state differences among the means. The models reporting OR and coefficients are comparing to a referent. In the text we state the differences among the other categories. For those results we used what in STATA is called lincom (linear combination of parameters which computes point estimates, SE etc for combination of coefficients after any estimation command

Table 4. ‘Ref’ is shortened. Need to define or write in full.

This has been corrected

Discussion : 231. Any suggestion as to the reason behind the differences? Did their methods differ subtly, or genetics, semen quality??

This has been addressed. A suggestion was made

This statement needs explaining further. Why do these results point to this? Is it not just because of numbers of sperm? Why quality? Explain.

This has been addressed. A suggestion was made

241-248 and 251-254. These sections say exactly the same thing. Either needs combining or remove the whole second para and find where to put the information in the first sentence (249+).

This suggestion has been accepted and applied.

References

Are the semicolons with the author names in the reference guide, I don’t remember them?

Semicolons are required

I believe that at the end of the reference Animal requires ‘Available online: www…. (accessed on ……….).’ for every reference

Did not see this in ‘Instructions to Authors’

Reference 2.Extra space at start

Corrected

Reference 3. Needs lining up with others

Corrected

Reference 9. Remove hyperlink from title

Corrected

Ref 11. IS there not more to this one? It seems incomplete?

Corrected

Ref 13. Everyone else has a bolded year this one doesn’t. Refer to the reference guide for authors as to which way this should be done.

Corrected

Reviewer 2 Report

Title: Combining fixed-time insemination and improved catheter design in an effort to improve swine reproduction efficiency

Authors: Matthew McBride; Rocio Amezcua; Glen Cassar; Terri O’Sullivan; Robert Friendship

Version: 1

Reviewer’s report:

This paper discusses methods to reduce semen dosage in pigs, whilst maintaining reproductive efficiency. I believe that there are several things that the authors need to address before it is considered for publication. 

Level of interest:  An article of some importance in its field.

Quality of written English: Good.

Statistical review: The manuscript does not need to be reviewed by a statistician.

Revisions:

 Simple summary:

Change about to approximately.

Single fixed-time artificial insemination needs to explained here.

Numbers 1-10 should be written as one-ten (e.g.: three billion; two hormone treatments; one of four treatment groups).

In general terms, the simple summary needs to made simpler – some of the terms used would not be understood by a non-specialist.

Abstract:

The abstract largely repeats the simple summary. If the simple summary is changed to be written in more layman’s terms, then the abstract is OK. Again, numbers 1-10 should be written as one-ten, as in the simple summary.

Introduction:

Generally clear introduction that describes the importance of the work. There are however some minor compulsory corrections that should be carried out:

Over-use of commas throughout.

Some of the language should be improved, for example, “performed” is better than “done” (Line 44).

Some of the introduction reads more like a thesis introduction rather than the introduction to a paper; terms such as “in a previous study” should be avoided.

The order of the introduction needs to be improved – it would be sensible to introduce AI before FTAI, and to give more generally background information to set the scene more effectively. The paragraph starting on Line 67 would be better placed earlier.

Again, numbers 1-10 should be written as one-ten.

Materials and Methods:

This section is written more like a thesis materials and methods section than a manuscript. The data and statistical analysis section needs to be re-written; stating that data were entered in Excel is not necessary.

Results: Generally satisfactory results section, but it would be better to present some of these data graphically rather than using four tables

 Discussion: This section is satisfactory, but more references should be referred to. The last sentence is vague (Line 266) and should definitely be deleted.

It would be useful to refer back to the specific Tables that are being referred to.

Again, numbers 1-10 should be written as one-ten.

In general, more references need to be used. Referring to only 16 articles in a peer-reviewed article is insufficient.

Author Response

This paper discusses methods to reduce semen dosage in pigs, whilst maintaining reproductive efficiency. I believe that there are several things that the authors need to address before it is considered for publication. 

Level of interest:  An article of some importance in its field.

Quality of written English: Good.

Statistical review: The manuscript does not need to be reviewed by a statistician.

Revisions:

 Simple summary:

Change about to approximately.

Re:This suggestion has been accepted and applied

Single fixed-time artificial insemination needs to explained here.

Re:Tried to make this a bit clearer, but the simple summary has a word limit and we have exceeded it.

Numbers 1-10 should be written as one-ten (e.g.: three billion; two hormone treatments; one of four treatment groups).

This suggestion has been accepted and applied

In general terms, the simple summary needs to made simpler – some of the terms used would not be understood by a non-specialist.

This was addressed

Abstract:

The abstract largely repeats the simple summary. If the simple summary is changed to be written in more layman’s terms, then the abstract is OK. Again, numbers 1-10 should be written as one-ten, as in the simple summary.

Small changes were made to address this

Introduction:

Generally clear introduction that describes the importance of the work. There are however some minor compulsory corrections that should be carried out:

Over-use of commas throughout.

Some commas removed where applicable

Some of the language should be improved, for example, “performed” is better than “done” (Line 44).

This suggestion has been accepted and applied

Some of the introduction reads more like a thesis introduction rather than the introduction to a paper; terms such as “in a previous study” should be avoided.

This suggestion has been accepted and applied

The order of the introduction needs to be improved – it would be sensible to introduce AI before FTAI, and to give more generally background information to set the scene more effectively. The paragraph starting on Line 67 would be better placed earlier.

This suggestion has been accepted and applied.  Second and third paragraphs have been switched

Again, numbers 1-10 should be written as one-ten.

This suggestion has been accepted and applied

Materials and Methods:

This section is written more like a thesis materials and methods section than a manuscript. The data and statistical analysis section needs to be re-written; stating that data were entered in Excel is not necessary

. This suggestion has been accepted and applied

Results: Generally satisfactory results section, but it would be better to present some of these data graphically rather than using four tables.

Table 2 was removed, so only 3 tables now

 Discussion: This section is satisfactory, but more references should be referred to. The last sentence is vague (Line 266) and should definitely be deleted.

Two more references have been added.

The last sentence has been replaced

It would be useful to refer back to the specific Tables that are being referred to.

Addressed

.In Discussion referring to tables in not common, that is done in the result section.

Again, numbers 1-10 should be written as one-ten.

This suggestion has been accepted and applied

In general, more references need to be used. Referring to only 16 articles in a peer-reviewed article is insufficient.

 Two more references have been added.

Round 2

Reviewer 1 Report

I believe that this paper has been improved for publication. 

My one comment is that the formatting (ie text size etc) seems to be different for many paragraphs and I don't believe that the tables are formatted correctly for the journal. 

The statistical concerns from my original review have been answered and all comments have been answered.